# Weaving the future: Maturity and key factors in women's cultural entrepreneurship in Colombia's artisan sector

Patricia Mendivil-Hernández[1], Jorge Alvis-Arrieta[2]*, Miguel Garcés-Prettel[3]

1 Regional and Local Development at the Technological University of Bolívar, Research lecturer at the Caribbean University Corporation CECAR, Colombia, 2 Economic and Territorial Integration and Development, Associate Professor at the School of Business, Law and Society and researcher at the Institute for Development, Economics and Sustainability Studies (IDEEAS) at the Technological University of Bolívar, Colombia, 3 Communication, Associate professor at the Technological University of Bolívar, Colombia

* jalvis@utb.edu.co

## Abstract

Women-led cultural enterprises in Colombia have emerged as a strategic opportunity to strengthen the cultural economy, support heritage tourism, and foster inclusive territorial development. This study examines the significant predictors of entrepreneurial maturity among women in the artisan sector. The analysis draws on data from 512 women registered in the 2023 database of the Departmental Institute of Culture and Tourism of Cundinamarca. A quantitative approach was employed, using a non-experimental, cross-sectional design and a binary logistic regression model. The findings reveal significant relationships between maturity and variables such as age, primary source of income, and participation in associations. Distance from the capital showed only a marginal effect. These results offer valuable insights for designing targeted strategies to enhance cultural and territorial development through the empowerment of women artisans.

## 1. Introduction

Women-led entrepreneurship has seen sustained growth in recent decades, establishing itself as a legitimate form of economic and social participation [1]. However, in many settings, women continue to face structural constraints that limit their access to resources, visibility, and business sustainability [1,2]. In Colombia, these limitations are partly rooted in historical contexts of instability and conflict, which have hindered recognition of women's productive labor, especially in activities that operate along the blurred boundary between economic work and domestic life [3,4].

In response to these conditions, many women have launched entrepreneurial initiatives that combine economic motivations with organizational forms grounded

**Data availability statement:** All relevant data are within the paper and its Supporting Information files.

**Funding:** The author(s) received no specific funding for this work.

**Competing interests:** The authors have declared that no competing interests exist.

in identity, community ties, and the use of traditional knowledge [5]. These ventures move beyond a purely instrumental logic and instead reveal a wide range of pathways in which entrepreneurship strengthens personal capabilities and helps consolidate sustainable productive units (Koellinger et al., 2013). As a result, research on women's entrepreneurship has gained traction not only for its economic implications but also for its contributions to social innovation, cultural valorization, and territorial development [6].

Within the cultural sphere, women-led artisan entrepreneurship has drawn particular interest for its ability to integrate tradition and innovation by transforming inherited knowledge into goods with both symbolic and market value [7,8]. This type of entrepreneurship allows us to observe how local cultural practices adapt to new demands while retaining key elements of identity that are meaningful to communities [8]. In this sense, artisan production contributes not only to economic livelihood but also to social cohesion and to the positioning of territories as sites of creative production and cultural tourism [9].

In Colombia, entrepreneurship has been promoted as a tool for economic revitalization, particularly in sectors such as culture, tourism, and innovation [24]. In departments like Cundinamarca, located in central Colombia and surrounding the capital city of Bogotá, urban and rural dynamics intersect under public policies that support entrepreneurship, including dedicated programme for the artisan sector. Yet despite these policy efforts, regional reports continue to reveal persistent gender gaps, especially in access to support networks, financing, visibility, and specialized training [10,11].

This coexistence of policy initiatives and structural inequality highlights the need to examine in great depth the factors that influence the sustainability of women's enterprises. Although the literature on entrepreneurship in Colombia is growing, empirical research on the determinants of organizational maturity in women-led initiatives remains limited, particularly within the artisan sector and in subnational contexts such as Cundinamarca.

This article seeks to identify the significant predictors of organizational maturity in women-led cultural enterprises in the artisan sector of Cundinamarca. To this end, we applied a binary logistic regression model to analyze associations between maturity and a set of sociodemographic, economic, organizational, and territorial variables. This is the first empirical study in the region to examine these dimensions jointly in order to better understand how organizational consolidation takes shape in such initiatives.

The findings offer relevant inputs for the design of public policies that are territorially rooted, culturally informed, and attentive to gender-based differences. The study is part of a doctoral research project within the PhD programme in Regional and Local Development at the Universidad Tecnológica de Bolívar in Cartagena, Colombia. Beyond its academic contribution, this work aims to support the design of institutional strengthening strategies by recognizing the role of women artisans as agents of transformation in local contexts shaped by persistent inequality.

## 2. Literature review

### 2.1. Women's entrepreneurship and social inequality

Women's entrepreneurship has gained visibility as a relevant factor for economic growth and social transformation across various contexts [1]. Women's participation in recent entrepreneurial ventures has increased steadily worldwide [12]. Nonetheless, pronounced gender inequalities persist, particularly in high-tech or more sophisticated sectors, where access to resources, support networks, and capital remains limited for many women [8,9].

From a demographic standpoint, women entrepreneurs are generally between the ages of 30 and 50 and tend to have a university education [13]. Yet, they often face barriers to accessing formal financing, leading many to launch their businesses using personal savings (Still, 2006). These limitations are even more acute in settings marked by high social vulnerability, where women contend not only with material deprivation but also with challenges such as functional illiteracy, structural poverty, and restrictive gender norms [14].

In many countries of the Global South, women's entrepreneurship emerges less as a deliberate career choice and more as a strategy for survival or improving living conditions [15]. In these cases, entrepreneurial projects often combine economic goals with social or community-oriented purposes, reflecting a logic distinct from conventional models that focus solely on profitability [15].

Several studies have identified gender-based differences in entrepreneurial motivation [16]. While men often prioritize financial objectives, women tend to integrate personal, cultural, or identity-related goals as the driving force behind their ventures [16,17]. Although women's entrepreneurship has historically received less scholarly attention than men's, its growing role in local and regional economies has led to increasing interest in the academic literature; still, significant gaps remain, especially regarding access to finance and strategic resources, which continue to function as structural barriers identified in numerous studies [18,19].

### 2.2. Women's entrepreneurship in Latin America and Colombia

Across Latin America, women's entrepreneurship has shown steady growth and a notable capacity to foster employment and innovation, even in environments where structural barriers remain deeply entrenched [1,3,9]. The women entrepreneurs face persistent challenges, including limited access to credit and the difficulty of balancing productive work with caregiving responsibilities [5]. Despite these constraints, the region reports one of the highest rates of female entrepreneurial activity in the world, at 17 percent [20], although its growth margins are narrower compared to those in other regions.

Several countries in the region, including Chile, Colombia, Costa Rica, Mexico, and Puerto Rico, have advanced public policies aimed at promoting entrepreneurship [3]. Nevertheless, outcomes remain uneven [21]. Structural gender disparities continue to manifest in labor force participation rates. In 2023, women's participation stood at 51.8 percent, while men's reached 74.4 percent, marking a gap of 22.6 percentage points [22]. This difference points to a historical disadvantage in women's access to the formal labor market and the sustainability of their economic initiatives.

Regional profiles of women entrepreneurs often reveal shared traits: they tend to be middle-aged, married, and mothers, with basic or intermediate levels of education [21,23]. Many run microenterprises with limited resources and minimal integration into commercial networks [23,24]. These conditions are further exacerbated by restricted access to financing, technology, information, and technical training, as well as by the burden of unpaid domestic labor [24].

In Colombia, women's entrepreneurship has gained visibility in recent decades and is increasingly recognized for its contribution to business dynamism [11]. Between 2008 and 2019, the female labor force participation rate rose from 46.6 to 53.1 percent, reflecting meaningful progress in women's economic inclusion [25]. This expansion has been accompanied by a more diverse entrepreneurial profile, with women entering the field from a wide range of social backgrounds, educational levels, and geographic regions. These shifts have contributed to a gradual transformation of Colombia's business ecosystem [26].

Since the 1990s, the country's entrepreneurial class has become increasingly heterogeneous, with greater variation in gender, class, and regional origin [27]. The expansion of educational opportunities, together with the growth of support networks and the implementation of gender equity policies, has facilitated women's integration into the entrepreneurial landscape [28,29]. In areas affected by armed conflict, women-led ventures have also played a role in rebuilding community life, generating income, and reweaving the social fabric [28–30].

## 2.3. Women's entrepreneurship in the artisan sector

Cultural entrepreneurship refers to those activities that involve the appropriation and creation of cultural meaning, both tangible and intangible, with the aim of producing goods and services that reflect and convey the values of a society, while also contributing to economic and social processes [31].

Within the artisan sector specifically, women's participation in cultural entrepreneurship plays a central role in local development. These initiatives not only strengthen women's economic autonomy but also contribute to the preservation of cultural traditions. Women entrepreneurs in this field have a positive effect on local economies by creating business models aligned with the principles of the creative economy [32]. In many cases, their work supports household economies and upholds social and cultural values through artisanal practices transmitted across generations [9]. As a result, artisan entrepreneurship serves both as a means of sustainable development and as a mechanism for safeguarding cultural heritage in connection with the diversification of tourism [33].

In Latin America, many women engage in artisan entrepreneurship out of necessity, often finding in it a way to integrate their productive and personal responsibilities more effectively [3]. Although many lack formal business training, they are often able to overcome structural and material challenges thanks to their motivation and the opportunities available within their communities. These experiences, in turn, contribute to building collective resilience [34].

The recent resurgence of the artisan sector has been shaped by renewed interest in local identity and cultural rootedness [34]. These dynamics have opened new spaces for women to take on increasingly prominent roles [34]. Women-led craft initiatives have made meaningful contributions to the economic development of local communities across diverse contexts [35]. By channeling creativity, skill, and entrepreneurial vision into their work, these women support community-based economies, create jobs, and help promote vocational education [35]. Artisan production provides them with a space to develop commercial capacities, showcase their work, and engage actively in the labor market, contributing to both social and economic development [13].

This type of entrepreneurship also serves as an active means of preserving intangible cultural heritage [8]. It sustains artisanal techniques, knowledge, and narratives passed down through generations, which are often incorporated into local tourism offerings. From this perspective, women's entrepreneurship in the artisan sector should be understood not only as an economic strategy, but also as a practice of cultural preservation with deep social and symbolic implications.

## 2.4. Maturity in entrepreneurship

Entrepreneurial maturity represents a critical stage within the organizational life cycle, marked by a substantial degree of both operational and strategic stability [36,37]. This phase is defined not only by a consolidated presence in the market, but also by internal efficiency, strengthened organizational structures, and the development of long-term capabilities that sustain competitiveness and innovation [36].

From an evolutionary perspective, the life cycle of a venture can be understood as a continuous process encompassing several stages, beginning with inception, followed by growth and maturity [37]. This trajectory can be analyzed through variables such as the age of the enterprise, its size, and sales volume [7]. These elements help classify organizations into different developmental phases, ranging from early efforts to stabilize operations to full market consolidation. A business is generally considered mature once it achieves financial stability and a strong market position, typically after more than ten years of operation. Ventures between one and four years old are in their initial phase, focused on defining activities

and achieving early growth. Those between five and nine years are considered to be in a growth stage, oriented toward expansion and consolidation [37].

Building on this perspective, the model that highlights several key factors in the maturation process, including the enterprise's age, organizational size, stability, exposure to disruptive change, and level of sectoral recognition [38]. This framework helps explain how ventures undergo structural adjustments before reaching maturity. Moreover, the concept of maturity extends beyond organizational attributes to include the entrepreneur's own professional trajectory. Older entrepreneurs often bring a valuable mix of experience, discernment, and resilience to the business environment. These mature profiles tend to benefit from targeted support programme such as continuous training, mentorship opportunities, and access to tailored financial resources [39].

Organizational maturity comprises multiple dimensions. It can be examined through perspectives such as digital maturity, knowledge management, enterprise architecture, and strategic alignment. Each of these dimensions reflects an organization's readiness to adapt to shifting environments and maintain a competitive edge [36]. Mature organizations typically focus on ensuring stability [37], strengthening internal processes, and integrating systems that support long-term operational consistency. However, maturity should not be understood as a static endpoint. Rather, it demands ongoing renewal, adaptability, and strategic leadership to remain relevant. At this stage, the core challenge is no longer survival, but the capacity to maintain a competitive, profitable, and resilient business model.

### 2.5. Entrepreneurial maturity from a gender perspective

The maturity of women-led enterprises is shaped by a progressive process of learning and the accumulation of experience over time. This process involves the continuous development of entrepreneurial skills and the incorporation of knowledge acquired throughout the entrepreneurial journey [40]. It also depends on the ability to manage available resources effectively, especially in sectors where competitiveness is high [41]

Several external factors also influence this process. Personal and professional support networks play a key role by offering guidance, technical assistance, access to funding, and strategic connections that contribute to business development [42]. In addition, access to tools for training, technology, and management is a decisive resource for strengthening productive initiatives [40].

Education and business training are particularly relevant. Instruction in leadership and organizational management enhances decision-making capacity and increases the likelihood of long-term sustainability [34]. Nonetheless, structural barriers persist that limit access to these resources, especially in rural areas or in sectors marked by economic precarity [43,44].

Association with others is considered one of the most important factors in the entrepreneurial maturity of women, as it enables them to collaborate in networks that foster mutual support in a business environment often defined by competition [33,39].

Based on this theoretical framework, and in response to the gaps identified in research on women's cultural entrepreneurship at the subnational level, the following hypotheses guide the empirical analysis of this study:

**H1.** The older the entrepreneur, the greater the likelihood that her enterprise has reached a level of maturity, as accumulated experience contributes to organizational consolidation [34,44]

**H2.** Entrepreneurs whose activity constitutes their primary source of income are more likely to reach organizational maturity, as this reflects a higher level of commitment, continuity, and prioritization [39,40].

**H3.** Membership in associations or networks is positively related to entrepreneurial maturity, as it facilitates access to resources, shared learning, and opportunities for organizational development [2,42].

### 3. Methodology

This study adopts a quantitative approach with a non-experimental, cross-sectional analytical design, focusing on the identification of relationships between variables that explain organizational maturity in women-led cultural enterprises. To

respond to the research objective, the analysis draws on the official database of the Instituto Departamental de Cultura y Turismo de Cundinamarca (IDECUT, 2023), which contains sociodemographic, operational, and geographic information on 512 women entrepreneurs in the artisanal sector. This dataset provides a representative snapshot of the female cultural entrepreneurship landscape in the region during the year under study.

The variables were selected based on their theoretical and operational relevance. The dependent variable was entrepreneurial maturity, operationalized as a binary outcome (mature/ not mature), following criteria defined by IDECUT and aligned with recognized standards of organizational consolidation. Independent variables included the entrepreneur's age (in years), membership in associations or collaborative networks, whether the enterprise served as the main source of income, the type of craft practiced (grouped into representative categories), and the geographic distance in kilometers between the entrepreneur's residence and Bogotá D.C., the nation's capital.

A binary logistic regression model was employed to estimate the probability that an enterprise would reach an organizational maturity stage, based on the selected independent variables. Binary logistic regression is appropriate when the dependent variable has two possible outcomes. This model estimates the relationship between a binary dependent variable and one or more independent variables using a logistic function that transforms a linear combination of predictors into a probability between 0 and 1 [45]. The model coefficients indicate how each independent variable influences the probability of the event occurring, and the results are interpreted in terms of odds ratios, which reflect the change in the probability of the event given a variation in the predictor variables [45].

The analysis was performed using SPSS software, version 29.0, and statistical significance was set at a threshold of $p < 0.05$. Model quality was assessed using Nagelkerke's pseudo-$R^2$, which allowed for an evaluation of its overall explanatory power. The results are presented and analyzed in the following section. Prior to conducting the statistical analysis, the technical quality of the database was verified according to IDECUT's standards (2023). While the variables were not derived from psychometric scales, exploratory analyses were conducted to assess the internal consistency of the data. Cronbach's alpha, applied to a subset of ordinal variables, yielded a general score of 0.85. For dichotomous variables, the Kuder-Richardson Formula 20 (KR-20) produced coefficients above 0.70. Although these values should be interpreted with caution due to the nature of the data, they indicate a reliable dataset with sufficient technical robustness for analytical purposes.

## 4. Results

This section presents the findings derived from the binary logistic regression model used to identify the determinants of organizational maturity in women-led cultural craft enterprises in Cundinamarca. Five independent variables were analyzed: age, membership in associations, the role of the enterprise as the main source of income, type of craft produced, and geographical distance to Bogotá D.C. The objective was to estimate how each factor predicts the likelihood that a venture has reached an advanced stage of organizational consolidation.

Table 1 summarizes the logarithmic coefficients (B), standard errors, significance levels (Sig.), and odds ratios (Exp(B)) corresponding to each predictor. These results reveal the most explanatory variables and provide the empirical foundation for validating the hypotheses formulated during the literature review.

As shown in the results, three variables display a statistically significant relationship with enterprise maturity: the entrepreneur's age (p = .001), the enterprise's role as the primary source of income (p = .001), and association membership (p = .021). Not only are these effects statistically robust, but their odds ratios are also greater than one, indicating a positive influence on the dependent variable. These findings empirically support the hypotheses regarding the role of structural factors in the organizational consolidation of women-led cultural enterprises.

The estimated theoretical model is graphically represented below, synthesizing the weight and statistical significance level of each predictor variable in relation to the dependent variable.

Additionally, Fig 1 displays the magnitude of the estimated effects through the odds ratios. The role of the enterprise as the main source of economic support exhibits the highest impact (OR = 2.66), followed by associativity (OR = 1.65) and

 

**Table 1. Results of the binary logistic regression model predicting the maturity of cultural craft enterprises in Cundinamarca.**

| Predictor | B | Standard Error | Wald | df | Sig. | Exp(B) |
|---|---|---|---|---|---|---|
| Age | 0.068 | 0.010 | 50.309 | 1 | .001 | 1.070 |
| Association Membership | 0.503 | 0.218 | 5.316 | 1 | .021 | 1.654 |
| Main Economic Activity | 0.770 | 0.225 | 18.911 | 1 | .001 | 2.657 |
| Craft Category (Model) | | | 8.197 | 4 | .085 | |
| M1: Weaving | −0.179 | 0.309 | 0.337 | 1 | .562 | 0.836 |
| M2: Costume Jewelry | 0.756 | 0.479 | 2.491 | 1 | .114 | 2.129 |
| M3: Basketry | −0.512 | 0.357 | 2.053 | 1 | .152 | 0.599 |
| M4: Leatherwork | −0.591 | 0.325 | 3.315 | 1 | .069 | 0.554 |
| Distance to Bogotá | 0.269 | 0.149 | 3.257 | 1 | .071 | 1.308 |
| Constant | −3.749 | 0.606 | 38.231 | 1 | <.001 | 0.024 |

*Note 1.* B coefficients and odds ratios (Exp(B)) were estimated using binary logistic regression. The dependent variable is enterprise maturity. Independent variables include age, association membership, main economic activity, craft category, and distance to the capital.

*Note 2.* Distance to the capital was measured in kilometers from the entrepreneur's place of residence to Bogotá D.C., the capital of Colombia and the region's primary market.

*Note 3.* Significance values (Sig.) indicate the statistical confidence level associated with each estimated effect in the model. Source: Own elaboration.

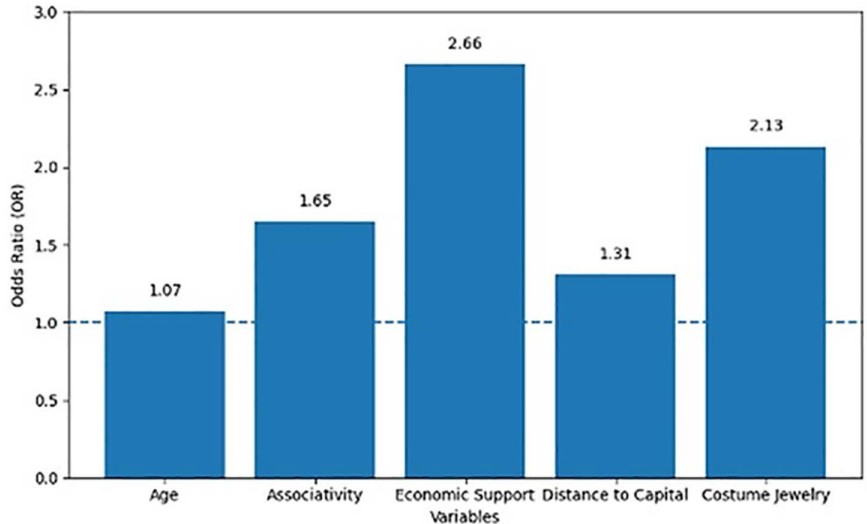

**Fig 1. Determinant Factors of Women's Entrepreneurial Maturity in the Craft Sector, by Significance Level and Odds Ratio (OR).** Note 6. Determinant factors of women's entrepreneurial maturity in the craft sector, according to the statistical model. Significance values are based on binary logistic regression analysis. Source: Own elaboration.

age (OR = 1.07), all with confirmed levels of statistical significance. These results indicate that enterprises which represent the primary income source, are led by older women, and are integrated into networks or associations, have a greater probability of reaching advanced stages of organizational maturity.

In contrast, geographical distance to the capital, Bogotá D.C., shows a positive but marginally significant effect (OR = 1.31; p = 0.071). Although the relationship is weak, it may suggest a relative advantage in terms of proximity to urban centers or major markets and access to institutional resources. Regarding the type of craft, the overall category set did not reach statistical significance (p = 0.085); however, the subcategory of jewelry, while not statistically significant (p = 0.114),

exhibited a high odds ratio (OR = 2.13), which may indicate a non-conclusive effect direction that warrants further exploration in future research.

Fig 2, in turn, presents the relative explanatory contribution of each variable within the model, based on the calculation of incremental pseudo-$R^2$. Primary economic sustenance emerges as the predictor with the greatest explanatory weight (39.7%), followed by age (23.8%) and associativity (15.9%). In contrast, the variables of distance to the capital (12.7%) and type of craft (7.9%) show a more limited contribution, which aligns with their lower levels of statistical association observed in the model.

These visualizations allow for a comprehensive interpretation of the findings: while Fig 3 identifies which factors significantly determine entrepreneurial maturity and to what extent, Fig 1 assesses their relative weight within the model. Together, they reinforce the empirical foundation of the study and provide a robust basis for theoretical and practical discussion in the context of women-led cultural enterprises.

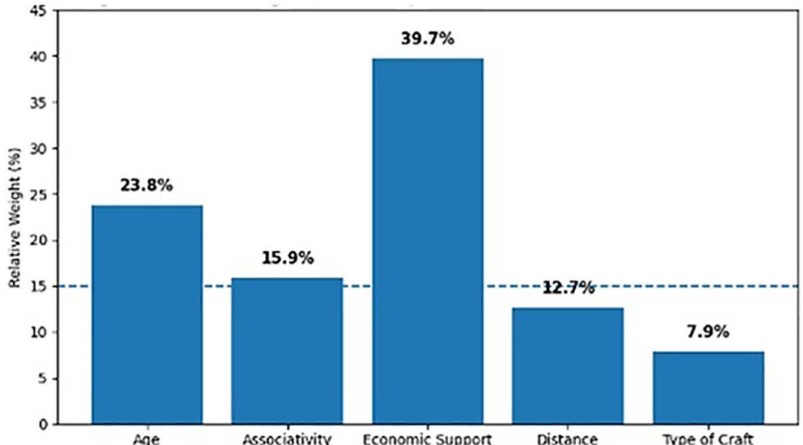

**Fig 2. Relative Weight of Each Independent Variable Within the Overall Model.** Note 7. The figure shows the relative weight (in percentages) of the independent variables that were statistically significant in the logistic regression model. The predictors with the greatest explanatory power were primary economic sustenance, age, and associativity. Source: Own elaboration.

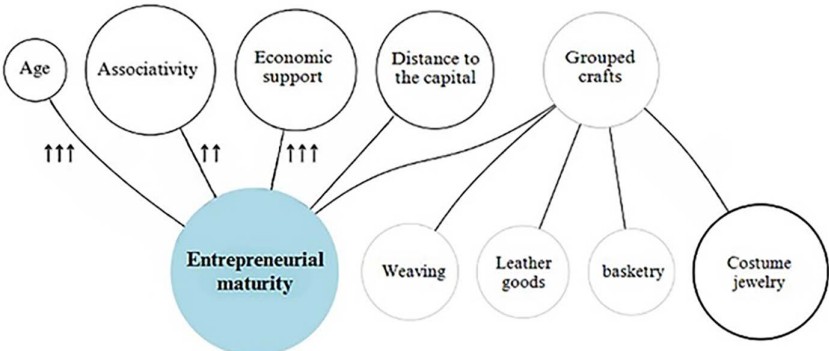

**Fig 3. Predictors of female entrepreneurship maturity in the artisanal sector in Cundinamarca. Note 4.** The diagram visualizes the independent variables associated with enterprise maturity (the dependent variable), represented with arrows according to their level of statistical significance. Bold circles denote each predictor of the dependent variable. Author's elaboration based on data from IDECUT (2023). **Note 5.** ↑↑↑ = highly significant ($p < 0.001$); ↑↑ = significant ($p < 0.05$); ↑ = marginal ($p < 0.10$); no arrow = not significant. Source: Own elaboration.

## 5. Discussion

The discussion section is organized according to the hypotheses tested, contrasting each empirical finding with existing literature and theoretical frameworks. Overall, the results confirm that the maturity of women-led cultural enterprises in the artisanal sector of Cundinamarca is shaped by a combination of individual, organizational, and territorial factors, thereby extending current approaches to entrepreneurial maturity in cultural and gender-based contexts.

First, the age of the entrepreneur showed a positive and significant association with the likelihood of maturity, which supports Hypothesis 1 and is consistent with studies that emphasize the value of accumulated experience for the sustainability of projects [34]. In the context studied, age reflects not only experience, but also a greater ability to build networks, manage resources, and adapt to the challenges specific to the artisanal sector. This finding expands existing theory by demonstrating that age operates not merely as a demographic variable, but as a proxy for relational capital and adaptive capacity in culturally embedded enterprises.

Hypothesis 2 was also confirmed, insofar as considering the enterprise as the primary source of economic livelihood significantly increases the probability of organizational maturity. This finding reaffirms that direct economic commitment, beyond complementary motivations, results in greater professionalization of the initiatives. As noted by Ruiz-Martínez et al. [32] and Sabater Fernández [40], when the enterprise constitutes the main source of income, women entrepreneurs tend to develop more solid structures, make strategic decisions more clearly, and actively seek financial stability. Our results extend this literature by showing that economic centrality is not only a motivational driver but a structural condition for organizational consolidation in peripheral cultural economies.

The third hypothesis was also empirically supported: associativity appears as a significant predictor of maturity; this relationship demonstrates the role of social capital as a strategic resource [42,44]. Participating in networks or associations not only allows for the sharing of knowledge and learning, but also facilitates access to technical support, visibility opportunities, and processes of collective legitimation. In sectors such as artisanal production, where economies of scale are limited, these forms of cooperation are key to organizational sustainability. This contribution reinforces and operationalizes social capital theory within the specific context of women-led artisanal enterprises, providing empirical evidence of its explanatory power in organizational maturity models.

In contrast, the variables of distance to the capital and type of craft did not show statistical significance, although they did exhibit trends that merit attention. In the case of geographic distance to Bogotá D.C., the marginally positive effect suggests that greater proximity to urban centers may offer relative advantages in terms of access to markets, institutional networks, and public support policies. This result is consistent with the views of Yadav et al. [13], who argue that geographic environments with dense infrastructure and institutional presence enhance the organizational capacities of enterprises in emerging markets.

Although the type of craft did not show a significant association as a block, the subcategory of jewelry presented a high odds ratio, though not conclusive ($p = 0.114$). This pattern may be interpreted as a sign that some activities, such as jewelry making, may have more favorable conditions for commercialization, innovation, or alignment with cultural consumption trends, which justify their inclusion in future analyses differentiated by subsector.

The overall interpretation of the model, supported by the incremental analyses of pseudo-$R^2$ (see Fig 1), reinforces the idea that maturity does not depend on a single factor, but rather on a hierarchical combination in which main economic support, age, and associativity together explain nearly 80 percent of the total variance of the model, which demonstrates their structural relevance. This multidimensional perspective is consistent with the arguments of Flores Jaén et al. [1] who emphasize that organizational maturity involves both personal capabilities and institutional environmental conditions.

From a territorial and gender-based perspective, these findings extend existing development theories by showing that maturity in women-led cultural enterprises transcends economic efficiency and incorporates symbolic, cultural, and political dimensions. As argued by Araque Geney et al. [9] women-led cultural enterprises not only stimulate local economies by complementing other sectors, such as tourism, but also strengthen social ties, re-signify collective identities,

and preserve patrimonial knowledge. In this sense, maturity transcends indicators of economic efficiency and acquires a symbolic, cultural, and political dimension [7,37].

Based on these findings, this study expands current implications for public policy and professional practice. Age and the economic centrality of the enterprise suggest the need for differentiated strategies according to stages in the entrepreneurial life cycle, while associativity underscores the relevance of strengthening territorial networks, peer-learning spaces, and collaborative platforms.

In peripheral regions, where distance from urban centers limits access to resources, such strategies can significantly help close structural gaps and promote conditions of equity in the organizational consolidation of women artisans. In line with the proposals of Castaño Castaño et al. [4], these policies should recognize the value of cultural entrepreneurship not only as an economic activity, but as a vehicle for empowerment, social cohesion, and territorial transformation led by women.

The results also allow us to foresee implications for the strengthening of community-based cultural tourism, particularly in contexts where women-led artisanal enterprises integrate identity practices, territorial ties, and economic sustainability. The organizational maturity observed in these initiatives may become a platform for developing cultural products with patrimonial and touristic value, capable of energizing local economies without uprooting their symbolic meanings.

## 6. Conclusion

The findings of this study show that the maturity of women-led cultural enterprises in the artisanal sector of Cundinamarca does not follow a single pattern but emerges from the interaction of life trajectories, economic commitment, and collective participation. Beyond being purely economic processes, these trajectories embody forms of sustainability that intertwine identities, traditional knowledge, and social relations in territories marked by structural inequalities.

This study extends existing theories of entrepreneurial maturity by demonstrating that maturity in cultural and gender-based enterprises cannot be reduced to economic performance alone, but must be understood as a multidimensional construct integrating social capital, territorial embeddedness, and symbolic value creation.

It is confirmed that variables such as age, main economic livelihood, and associativity not only predict organizational maturity but also reflect structural conditions that must be considered in the design of public policies. Likewise, the lower probability of consolidation in areas farther from the capital highlights the need to address spatial gaps that affect access to resources, markets, and institutional infrastructure. These results advance current theoretical models by empirically validating the role of relational and territorial factors as central components of organizational consolidation in peripheral cultural economies.

In this regard, the findings contribute to theory by linking organizational maturity with processes of cultural reproduction and territorial sustainability, positioning women-led artisanal enterprises as hybrid actors that simultaneously operate within economic, cultural, and social systems. Organizational maturity, understood as a multidimensional process, can become a platform for connecting these enterprises with creative economic circuits and community-based cultural tourism. This theoretical integration bridges entrepreneurship studies with cultural economics and gender-based development frameworks, offering a more comprehensive explanation of how maturity emerges in culturally embedded enterprises.

An important limitation of the study lies in the fact that, although relevant predictive factors were identified, other dimensions such as access to digital skills, marketing channels, or the impact of existing cultural policies were not addressed. Therefore, it is recommended that future research in this line expand the analysis to other sectors of the cultural ecosystem, incorporate variables related to the institutional environment, and further explore the territorial dynamics that enhance or constrain the organizational development of women artisans in local contexts. Future research should also test the proposed multidimensional maturity framework in other regional and cultural contexts in order to refine and generalize its theoretical contributions.

## Limitations and future research

This study presents certain limitations that should be acknowledged. First, the cross-sectional design restricts causal inference. Second, variables related to tourism, innovation processes, and institutional governance were not directly measured. Future research should incorporate longitudinal designs and differentiated subsector analyses in order to refine the understanding of maturity dynamics.

Additionally, future studies should explore how cultural entrepreneurship interacts with gendered power structures and regional development policies across different territorial contexts. Comparative studies across Colombian regions with different geographical and institutional conditions, such as Cauca or Vaupés, may also provide valuable insights into how territorial factors influence the maturity of women-led artisan enterprises.

## Supporting information

**S1 File. Database including grouped age and distance in kilometers.**
(XLSX)

## Acknowledgments

I would like to thank the Technological University of Bolivar (Colombia) for all its support, as well as the Colombian Ministry of Science, Technology and Innovation through Call No. 22, and also the team of lecturers on the Doctorate programme in Regional and Local Development at the UTB. The authors also thank the anonymous reviewers for their valuable comments and suggestions, which helped improve the manuscript.

## Author contributions

**Conceptualization:** Jorge Alvis-Arrieta, Patricia Mendivil-Hernández, Miguel Garcés-Prettel.

**Data curation:** Jorge Alvis-Arrieta, Patricia Mendivil-Hernández, Miguel Garcés-Prettel.

**Formal analysis:** Jorge Alvis-Arrieta, Patricia Mendivil-Hernández, Miguel Garcés-Prettel.

**Investigation:** Patricia Mendivil-Hernández, Miguel Garcés-Prettel.

**Methodology:** Jorge Alvis-Arrieta, Patricia Mendivil-Hernández, Miguel Garcés-Prettel.

**Project administration:** Patricia Mendivil-Hernández.

**Software:** Patricia Mendivil-Hernández, Miguel Garcés-Prettel.

**Supervision:** Miguel Garcés-Prettel.

**Validation:** Patricia Mendivil-Hernández, Miguel Garcés-Prettel.

**Writing – original draft:** Jorge Alvis-Arrieta.

**Writing – review & editing:** Jorge Alvis-Arrieta, Patricia Mendivil-Hernández, Miguel Garcés-Prettel.

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
