## [Decision Letter · Decision Letter 0]

2 Feb 2026

Dear Dr. Alvis-Arrieta,

plosone@plos.org. . . . A letter that responds to each point raised by the academic editor and reviewer(s). You should upload this letter as a separate file labeled 'Response to Reviewers'.A marked-up copy of your manuscript that highlights changes made to the original version. You should upload this as a separate file labeled 'Revised Manuscript with Track Changes'.An unmarked version of your revised paper without tracked changes. You should upload this as a separate file labeled 'Manuscript'.

We look forward to receiving your revised manuscript.

Kind regards,

Rafael Galvão de Almeida, PhD.

Academic Editor

PLOS One

Journal Requirements:

2. We note that your Data Availability Statement is currently as follows:

“The data are all contained within the manuscript and/or Supporting Information files, enter the following: All relevant data are within the manuscript and its Supporting Information files.”

3.  We note you have included a table to which you do not refer in the text of your manuscript. Please ensure that you refer to Table 1 in your text; if accepted, production will need this reference to link the reader to the Table.

**Additional Editor Comments:**

Please, adress all points raised by the reviewers, especially in matters of organization and references.

Reviewers' comments:

Reviewer's Responses to Questions

**Comments to the Author**

1. Is the manuscript technically sound, and do the data support the conclusions?

Reviewer #1: Yes

Reviewer #2: Yes

2. Has the statistical analysis been performed appropriately and rigorously?

Reviewer #1: I Don't Know

Reviewer #2: Yes

3. Have the authors made all data underlying the findings in their manuscript fully available?

Reviewer #1: Yes

Reviewer #2: No

4. Is the manuscript presented in an intelligible fashion and written in standard English?

Reviewer #1: Yes

Reviewer #2: Yes

Reviewer #1: The central claim of this paper is that there is evidence correlating specific factors with the maturity of women-led artisan enterprises in Cundinamarca, Colombia. Statistically significant factors include the entrepreneur’s age, whether the enterprise is the primary source of income, and the entrepreneur’s level of participation in professional associations. This paper makes a valuable contribution to the literature on the Latin American artisan sector. Although the dataset focuses narrowly on Cundinamarca, the region’s proximity to Colombia’s capital positions it as a useful case study for craft communities near major urban centers. Given the scarcity of data-driven research on artisan enterprises, this paper—with minor revisions—will be of interest to researchers, policymakers, and practitioners in Latin America, the United States, and other majority-world contexts.

The authors effectively situate their research within broader scholarship on women’s entrepreneurship, with specific attention to Latin America, Colombia, and the artisan sector. I particularly appreciate the inclusion of Colombian scholars alongside international sources. That said, the References section requires attention. Several entries need copyediting (e.g., spacing before punctuation). Reference #11 contains a double period and lacks publisher information and a URL, despite the text being available online. Reference #1 incorrectly lists a 2026 publication date; the correct date appears to be 2023. Reference #8 is missing page numbers.

While econometrics is not my area of expertise, the data and conclusions are clearly presented and accessible beyond quantitative fields. For future research, I encourage comparative analyses across Colombian regions. The finding that distance from the capital is insignificant is intriguing and may reflect the geographic scope of the study. A comparison with regions such as Cauca or Vaupés could yield different conclusions.

Reviewer #2: The following issues need to be addressed:

1. In the introduction section, the sentence beginning "Women led entrepreneurship..." requires a supporting academic reference.

2. The sentence beginning "This type of entrepreneurship..." requires a supporting academic reference.

3. The sentence beginning "In Colombia, entrepreneurship has been prompted....." requires a supporting academic reference.

4. Please ensure that the manuscript is written in UK English rather than US. Please edit all instances. for example "programme" and not "program".

5. In Section 2, the sentence beginning "Women's entrepreneurship has...." requires a supporting academic reference.

6. The sentence beginning "Nonetheless, pronounced gender..." requires a supporting academic reference.

7. The sentence beginning "In these cases..." requires a supporting academic reference.

8. The sentence beginning "Several studies have identified..." requires a supporting academic reference.

9. In section 2.2 the sentence beginning "Across Latin America..." requires a supporting academic reference.

10. The sentence beginning "The women’s entrepreneurs..." requires a supporting academic reference.

11. In section 2.2, the countries in the sentence beginning "Several countries in the region..." needs to be put into alphabetical order.

12. In section 2.2, the sentence beginning "Regional profiles of...." requires a supporting academic reference.

13. In section 2.2, the sentence beginning "These conditions are further..." requires a supporting academic reference.

14. In section 2.2, the sentence beginning "In Colombia women's..." requires a supporting academic reference.

15. In section 2.2, the sentence beginning "The expansion of educational opportunities..." requires a supporting academic reference.

16. In section 2.3, the sentence beginning "In Latin America, many..." requires a supporting academic reference.

17. In section 2.3, the sentence beginning "The recent resurgence..." requires a supporting academic reference.

18. In section 2.3, the sentence beginning "Women led craft initiatives..." requires a supporting academic reference.

19. In section 2.3, the sentence beginning "This type of entrepreneurship...." requires a supporting academic reference.

20. In section 2.4. the sentene beginning "Entrepreneurial maturity represents..." requires a supporting academic reference.

21. In section 2.4, the sentence beginning "From an evolutionary perspective..." requires a supporting academic reference.

22. In section 2.4 the sentence beginning "Mature organizations typically..." requires a supporting academic reference.

23. The methodology section is a little brief and would benefit from some further academic underpinning to support the approach undertaken.

24. Figure 2 requires both x and y axis labels. I would put the OR amount on the graph rather than under the label.

25. Figure 3 also requires both X and Y axis labels.

26. I would reorganise the discussion and Conclusions sections. The Discussion section to clearly answer each hypothesis in contrast to the existing literature. Further develop the conclusions section to confirm how this study extends existing theory. Please directly identify how existing literature is further development Thereafter, a section on how this study extends existing implications for policy and practice would be useful followed by study limitations and further research opportunities.

.

Reviewer #1: **Yes:**Cynthia Lawson JaramilloCynthia Lawson JaramilloCynthia Lawson JaramilloCynthia Lawson Jaramillo

Reviewer #2: No

---

## [Author Response · Author response to Decision Letter 1]

24 Mar 2026

Colombia, 17 March

Dear Editor and Reviewers,

We would like to express our sincere gratitude for the time and effort you have devoted to reviewing our manuscript, as well as for the valuable comments and suggestions you have provided. All your comments have been carefully considered and have contributed significantly to improving the scientific, methodological and conceptual quality of the work.

Based on your recommendations, we have revised the manuscript, incorporating the requested corrections and strengthening the theoretical framework, the methodology section, the presentation of results, the discussion and the conclusions. Below, we present a detailed and structured response to each of the comments made by the reviewers.

Reviewer Comment Response

Reviewer 1 Inclusion of the reference to Table 1 in the text. Thank you for your valuable suggestions. We have carried out the explicit inclusion of the reference to Table 1 within the body of the manuscript in order to ensure coherence between the text and the graphical elements (page 14).

Reviewer 1 General review of the reference list. Thank you for your valuable suggestions. We have carried out a thorough review of the reference list, correcting errors in formatting, punctuation, and consistency in accordance with the journal’s editorial standards.

Reviewer 1 Reference No. 11 contained a double full stop and lacked editor and URL information. Thank you for your valuable suggestions. We have carried out the correction of Reference No. 11 by removing the double full stop and incorporating the missing editor information and corresponding URL (page 25).

Reviewer 1 Reference No. 1 incorrectly indicated the year 2026; Reference No. 8 lacked page numbers. Thank you for your valuable suggestions. We have carried out the correction of the publication year of Reference No. 1 (2023). Regarding the missing page numbers in Reference No. 8, it is clarified that this source does not include page numbering due to its format on the virtual platform where it is hosted. Additionally, we have carried out the correction of spacing before punctuation across all references (page 26).

Reviewer 1 For future research, I encourage comparative analyses across Colombian regions. A comparison with regions such as Cauca or Vaupés could yield different conclusions. Thank you for this valuable suggestion. We have incorporated this recommendation in the “Limitations and future research” section of the manuscript by highlighting the relevance of comparative analyses across Colombian regions with different territorial conditions, including regions such as Cauca or Vaupés (page 23).

Reviewer 2 In Section 2.2, the countries in the sentence beginning “Several countries in the region…” need to be put into alphabetical order. Thank you for this observation. The list of countries has been reordered alphabetically as suggested. The sentence now reads: “Several countries in the region, including Chile, Colombia, Costa Rica, Mexico, and Puerto Rico, have advanced public policies aimed at promoting entrepreneurship (3).” This change has been incorporated in Section 2.2 of the revised manuscript (page 6).

Reviewer 2 Add an academic reference to the sentence “Women-led entrepreneurship…” (Introduction). Thank you for your valuable suggestions. We have carried out the incorporation of updated academic references that support this statement in the Introduction section (page 2).

Reviewer 2 Add an academic reference to the sentence “This type of entrepreneurship…”. Thank you for your valuable suggestions. We have carried out the inclusion of relevant theoretical references to support this statement (page 3).

Reviewer 2 Add an academic reference to the sentence “In Colombia, entrepreneurship has been promoted…”. Thank you for your valuable suggestions. We have carried out the inclusion of academic sources that support this statement (page 3).

Reviewer 2 Please ensure that the manuscript is written in UK English rather than US. Please edit all instances. for example "programme" and not "program".

Thank you for your valuable suggestions. We have carried out a comprehensive linguistic revision of the manuscript and adjusted it to British English standards, ensuring the consistent use of terms such as “programme” (page 4).

Reviewer 2 Section 2: “Women’s entrepreneurship has…” requires an academic reference. Thank you for your valuable suggestions. We have carried out the inclusion of the corresponding academic reference supporting this statement (page 5).

Reviewer 2 “However, the marked gender…” requires an academic reference. Thank you for your valuable suggestions. We have carried out the incorporation of relevant academic literature supporting this statement (page 5).

Reviewer 2 “In these cases…” requires an academic reference. Thank you for your valuable suggestions. We have carried out the inclusion of academic sources that support this argument (page 5).

Reviewer 2 “Several studies have identified…” requires an academic reference. Thank you for your valuable suggestions. We have carried out the inclusion of relevant references that strengthen the theoretical underpinning of this statement (page 5).

Reviewer 2 Section 2.2: “Throughout Latin America…” requires an academic reference. Thank you for your valuable suggestions. We have carried out the incorporation of relevant regional references supporting this statement (page 6).

Reviewer 2 “Women entrepreneurs…” requires an academic reference. Thank you for your valuable suggestions. We have carried out the inclusion of academic references on women’s entrepreneurship to support this statement (page 6).

Reviewer 2 Alphabetise the countries in “Several countries in the region…”. Thank you for your valuable suggestions. We have carried out the reorganisation of the countries into alphabetical order as requested (page 6).

Reviewer 2 “Regional profiles of…” requires an academic reference. Thank you for your valuable suggestions. We have carried out the inclusion of supporting bibliographic references for this statement (page 6).

Reviewer 2 “These conditions are even more…” requires an academic reference. Thank you for your valuable suggestions. We have carried out the incorporation of relevant academic sources supporting this argument (page 7).

Reviewer 2 “In Colombia, women…” requires an academic reference. Thank you for your valuable suggestions. We have carried out the inclusion of relevant national references supporting this statement (page 7).

Reviewer 2 “The expansion of educational opportunities…” requires an academic reference. Thank you for your valuable suggestions. We have carried out the incorporation of academic references supporting this statement (page 7).

Reviewer 2 Section 2.3: “In Latin America, many…” requires an academic reference. Thank you for your valuable suggestions. We have carried out the inclusion of relevant theoretical references supporting this statement (page 8).

Reviewer 2 “The recent resurgence…” requires an academic reference. Thank you for your valuable suggestions. We have carried out the incorporation of academic sources supporting this statement (page 8).

Reviewer 2 “Women-led craft initiatives…” requires an academic reference. Thank you for your valuable suggestions. We have carried out the inclusion of specific references on women’s craft entrepreneurship (page 8).

Reviewer 2 “This type of entrepreneurship…” requires an academic reference. Thank you for your valuable suggestions. We have carried out the incorporation of relevant references supporting this statement (page 8).

Reviewer 2 Section 2.4: “Entrepreneurial maturity represents…” requires an academic reference. Thank you for your valuable suggestions. We have carried out the inclusion of theoretical references on organisational maturity supporting this statement (page 9).

Reviewer 2 “From an evolutionary perspective…” requires an academic reference. Thank you for your valuable suggestions. We have carried out the incorporation of relevant references supporting this perspective (page 9).

Reviewer 2 “Mature organisations tend to…” requires an academic reference. Thank you for your valuable suggestions. We have carried out the inclusion of relevant academic references supporting this statement (page 10).

Reviewer 2 The methodology section is a little brief and would benefit from further academic underpinning. Thank you for your valuable suggestions. We have carried out the expansion of the methodology section by incorporating stronger academic underpinning for the adopted approach. Specifically, we have added a clearer explanation of the binary logistic regression model and included an additional methodological reference supporting the analytical procedure (page 12-13).

Reviewer 2 Figure 2 requires axis labels and OR values on the bars. Thank you for your valuable suggestions. We have carried out the modification of Figure 2 by incorporating labels on both axes and displaying the Odds Ratio (OR) values on each bar to improve interpretation (page 15).

Reviewer 2 Figure 3 requires axis labels. Thank you for your valuable suggestions. We have carried out the update of Figure 3 by including complete labels on the X and Y axes, as well as the corresponding percentage values (page 16).

Reviewer 2 Reorganise Discussion and Conclusions; expand theoretical contributions; include implications, limitations, and future research. Thank you for your valuable suggestions. We have carried out the reorganisation of the Discussion and Conclusions sections. The discussion has been structured to address each hypothesis by contrasting it with the existing literature, and implications for public policy and practice have been included. Additionally, we have carried out the expansion of the conclusions to highlight the theoretical contributions of the study, and we have incorporated limitations and future lines of research (page 19-25).

We are confident that the amendments made adequately address each of the comments raised and contribute to substantially improving the clarity, academic rigour, and theoretical and practical contribution of the manuscript.

We would like to reiterate our gratitude to the Academic Editor and the reviewers for their constructive comments, which have helped to strengthen this work. We remain open to any further comments they may deem relevant.

Yours faithfully,

The authors.

---

## [Editor Report · Decision Letter 1]

25 Mar 2026

WEAVING THE FUTURE: MATURITY AND KEY FACTORS IN WOMEN’S CULTURAL ENTREPRENEURSHIP IN COLOMBIA’S ARTISAN SECTOR

PONE-D-25-67330R1

Dear Dr. Alvis-Arrieta,

We’re pleased to inform you that your manuscript has been judged scientifically suitable for publication and will be formally accepted for publication once it meets all outstanding technical requirements.

Kind regards,

Rafael Galvão de Almeida, PhD.

Academic Editor

PLOS One

Additional Editor Comments (optional):

Please, add your reviewers in the Acknowledgements section.
---

## [Editor Report · Acceptance letter]

PONE-D-25-67330R1

PLOS One

Dear Dr. Alvis-Arrieta,

I'm pleased to inform you that your manuscript has been deemed suitable for publication in PLOS One. Congratulations! Your manuscript is now being handed over to our production team.

Kind regards,

on behalf of

Dr. Rafael Galvão de Almeida

Academic Editor

PLOS One